# GroSS Decomposition: Group-Size Series Decomposition for Whole Search-Space Training

## Abstract

We present Group-size Series (GroSS) decomposition, a mathematical formulation of tensor factorisation into a series of approximations of increasing rank terms. GroSS allows for dynamic and differentiable selection of factorisation rank, which is analogous to a grouped convolution. Therefore, to the best of our knowledge, GroSS is the first method to simultaneously train differing numbers of groups within a single layer, as well as all possible combinations between layers. In doing so, GroSS trains an entire grouped convolution architecture search-space concurrently. We demonstrate this with a proof-of-concept exhaustive architecure search with a performance objective. GroSS represents a significant step towards liberating network architecture search from the burden of training and finetuning.

## 1 Introduction

In recent years, there has been a flurry of Deep neural networks (DNNs) producing remarkable results on a broad variety of tasks. Generally, these methods have required careful network design, often relying on domain knowledge to design a structure which can encapsulate the task at hand. Neural Architecture Search (NAS) has provided an alternative to hand designed networks, allowing for the search and even direct optimisation of the network's structure.

The search space for architectures is often vast, with potentially limitless design choices. Furthermore, each configuration must undergo some training or finetuning for its efficacy to be determined. This has lead to the development of methods which lump multiple design parameters together, to reduce the search space in a principled manner (Tan & Le, 2019). As well as creating the need for sophisticated search algorithms (Liu et al., 2018; Wu et al., 2019), which can more quickly converge to an improved design. Both techniques reduce the number of search iterations and ultimately reduce the number of required training/finetuning stages.

Architecture search has so far, to the best of our knowledge, avoided exploring grouped convolution design. However, grouped convolution network design presents itself as an ideal candidate for architecture search. It has been been widely used particularly in some prevalent networks. ResNeXt (Xie et al., 2017) used grouped convolution for improved accuracy over the analogous ResNets (He et al., 2016). On the other hand, MobileNet (Howard et al., 2017) and various others (Zhang et al., 2018; Sandler et al., 2018) have utilised grouped convolutions in the depthwise case in a ResNet-style for extremely low-cost inference. With these architectures, grouped convolution has proven to be a valuable design tool for high-performance and low-cost design alike. Applying it for these contrasting performance profiles requires an intuition, which so far has remained relatively unexplored.

However, grouped convolution design implications have remained relatively unexplored. Decomposition of networks is time consuming. Also, there isn't necessarily a heuristic or intuition for how combinations of grouped convolutions with varying numbers of groups interact in a network. We tackle this in this work with the introduction of a Group-size Series (GroSS) decomposition. GroSS allows us to train the entire search space of architectures *simultaneously*. In doing so, we shift the expense of architecture search with respect to group-size away from decomposition and training, and towards cheaper test-time sampling. This allows for the exploration of possible configurations, while significantly reducing the need for imparting bias on the group design hyperparameter selection.

The contributions of this paper can be summarised as follows:

1. We present GroSS Decomposition – a novel formulation of tensor decomposition as a series of rank approximations. This provides a mathematical basis for grouped convolution as a series of increasing rank terms.

2. GroSS provides the apparatus for differentiably switching between grouped convolution ranks. Therefore, to the best of our knowledge, it is the first simultaneous training of differing numbers of groups within a single layer, as well as the all possible configurations between layers. Effectively training an entire architecture search space at once.

3. We explore this concurrently trained architecture space with a proof-of-concept exhaustive search. Illustrating the efficacy of the GroSS, as well as taking a step towards removing the train burden from architecture search.

## 2  RELATED WORK

Grouped convolution has had a wide impact on neural network architectures, particularly due to its efficiency. It was first introduced in AlexNet (Krizhevsky et al., 2012) as an aid for the single network to be trained over multiple GPUs. Since then, it has had a wide impact on DNN architecture design. ResNeXt (Xie et al., 2017) used grouped convolutions synonymously with concept of *cardinality*, ultimately exploiting the efficiency of grouped convolutions for high-accuracy network design. The reduced complexity of grouped convolution allowed for ResNeXt to incorporate deeper layers within the ResNet-analogous residual blocks (He et al., 2016). In all, this allowed higher accuracy with a similar inference cost as an equivalent ResNet. The efficiency of grouped convolution has also lead to several low-cost network designs. MobileNet (Howard et al., 2017) utilised a ResNet-like bottleneck design with depthwise convolutions–a special case of grouped convolutions where the number of groups is set to equal the number of in channels–for an extremely efficient network with mobile applications in mind. ShuffleNet (Zhang et al., 2018) was also based on a depthwise bottleneck, however, pointwise layers were also made grouped convolutions.

Previous works (Jaderberg et al., 2014; Denton et al., 2014; Lebedev et al., 2014; Vanhoucke et al., 2011) have applied low-rank approximation of convolution for network compression and acceleration. Block Term Decomposition (BTD) (De Lathauwer, 2008) has recently been applied to the task of network factorisation (Chen et al., 2018), where it was shown that the BTD factorisation of a convolutional weight was equivalent to a grouped convolution within a bottleneck architecture. Wang et al. (2018) applied this equivalency for network acceleration. Since decomposition is costly, these methods have relied on heuristics and intuition to set hyperparameters such as the rank of successive layers within the decomposition. In this paper, we present a method for decomposition which allows for exploration of the decomposition hyperparameters and all the combinations.

Existing architecture search methods have overwhelmingly favoured reinforcement learning. Examples of this include, but are not limited to, NASNet (Zoph et al., 2018), MNasNet (Tan et al., 2019), ReLeq-Net (Elthakeb et al., 2018). In broad terms, these methods all set a baseline structure, which is manipulated by a separate controller. The controller optimises the structure through and objective based on network performance. There has also been work in differentiable architecture search (Wu et al., 2019; Liu et al., 2018) which makes the network architecture manipulations themselves differentiable. In addition, work such as (Tan & Le, 2019) aims to limit the network scaling within a performance envelope to a single parameter.

These methods all have a commonality: the cost of re-training or finetuning at each stage motivates the recovery of the optimal architecture in as few training steps as possible, whether this is achieved through a trained controller, direct optimisation or significantly reducing the search space. In this work, however, we produce a method where the entire space is trained at once and therefore shift the burden of architecture search away from training.

## 3  METHOD

In this section, we will first introduce Block Term Decomposition (BTD) and detail how its factorisation can be applied to a convolutional layer. After that, we will introduce GroSS decomposition, where we formulate a unification of a series of ranked decompositions so that they can dynamically and differentially be combined. We detail the training strategy for training the whole series at

once. We describe our response reconstruction formulation to improve the approximation provided by factorisation.

## 3.1 GENERAL BLOCK TERM DECOMPOSITION

Block Term Decomposition (BTD) (De Lathauwer, 2008) aims to factorise a tensor into the sum of multiple low rank-Tuckers (Tucker, 1966). That is, given an $N^{th}$ order tensor $\mathbf{X} \in \mathbb{R}^{d_1 \times d_2 \times ... \times d_N}$, BTD factorises $\mathbf{X}$ into the sum of rank $(d'_1, d'_2, ..., d'_N)$ terms:

$$\mathbf{X} = \sum_{r=1}^{R} \mathbf{G}_r \times_1 \mathbf{A}_r^{(1)} \times_2 \mathbf{A}_r^{(2)} \times_3 ... \times_N \mathbf{A}_r^{(N)}$$

$$\text{where} \begin{cases} \mathbf{G} \in \mathbb{R}^{d'_1 \times d'_2 \times ... \times d'_N} \\ \mathbf{A}_r^{(n)} \in \mathbb{R}^{d_n \times d'_n}, n \in \{1, ..., N\} \end{cases} \tag{1}$$

In the above, $\mathbf{G}$ is known as the *core* and we will refer to $\mathbf{A}$ as *factors*. We use $\times_n$ to represent the *mode-n* product (De Lathauwer, 2008).

## 3.2 SINGLE CONVOLUTION TO BOTTLENECK WITH BTD

The weights of a convolution can be formulated as a 4th order tensor $\mathbf{X} \in \mathbb{R}^{d_1 \times d_2 \times d_3 \times d_4}$, where $d_1$ and $d_2$ represent the number of input and output channels, and with $d_3$ and $d_4$ being the spatial size of the filter kernel.

For convenience and clarity, we will simplify notation from the most general BTD to the 4D case as follows: $\mathbf{X} \in \mathbb{R}^{t \times u \times v \times w}$, with $\mathbf{B} = \mathbf{A}^{(1)}$ and $\mathbf{C} = \mathbf{A}^{(2)}$. Typically the spatial extent of each filter is small and thus we only factorise along $t$ and $u$, so that BTD for convolutional weights is expressed as follows:

$$\mathbf{X} = \sum_{r=1}^{R} \mathbf{G}_r \times_1 \mathbf{B}_r \times_2 \mathbf{C}_r$$

$$\text{where} \begin{cases} \mathbf{G} \in \mathbb{R}^{t' \times u' \times v \times w} \\ \mathbf{B} \in \mathbb{R}^{t \times t'} \\ \mathbf{C} \in \mathbb{R}^{u \times u'} \end{cases} \tag{2}$$

It can be shown that this factorisation of the convolutional weights forms a three-layer, bottleneck-style structure (Yunpeng et al., 2017): a pointwise ($1 \times 1$) convolution $\mathbf{P} \in \mathbb{R}^{t \times t' \times 1 \times 1}$, formed from factor $\mathbf{C}$; followed by a grouped convolution $\mathbf{R} \in \mathbb{R}^{u' \times t' \times v \times w}$, formed from core $\mathbf{G}$; and finally another pointwise convolution $\mathbf{Q} \in \mathbb{R}^{u' \times u \times 1 \times 1}$, formed from factor $\mathbf{B}$. With careful selection of the BTD parameters, the bottleneck approximation can be applied to any standard convolutional layer.

$$\begin{cases} R & = \text{Number of groups in the grouped convolution} \\ t & = \text{Number of input channels} \\ u & = \text{Number of output channels} \\ ot', u' & = \dfrac{t}{R} = \text{Group-size} \end{cases} \tag{3}$$

In Table 1, we detail how the dimensions of the bottleneck architecture compared to its corresponding convolutional layer. We also include how properties such as stride, padding and bias are applied within the bottleneck for equivalency with the original layer. It is worth noting that we often refer to the quantities $t'$ and $u'$ as the group-size. This quantity determines the number of channels present in each group and is equivalent to the rank of the decomposition.

## 3.3 GROUP-SIZE SERIES DECOMPOSITION

Group-size Series (GroSS) decomposition unifies multiple ranks of BTD factorisations. This is achieved by defining each successive factorisation relative to the lower order ranks. Thus we ensure that higher rank decompositions only contain information that was missed by the lower order

| | Filter Size | $C_{in}$ | $C_{out}$ | Groups | Bias | Stride | Padding |
|---|---|---|---|---|---|---|---|
| Original | $d_3 \times d_4$ | $d_1$ | $d_2$ | 1 | $B$ | $S$ | $P$ |
| | $1 \times 1$ | $d_1$ | $d_1$ | 1 | - | $S$ | $P$ |
| Decomposed | $d_3 \times d_4$ | $d_1$ | $d_1$ | $R$ | - | 1 | 0 |
| | $1 \times 1$ | $d_1$ | $d_2$ | 1 | $B$ | 1 | 0 |

Table 1: Convolution to grouped bottleneck. We detail how the convolutional parameters can be applied to the bottleneck when factorising using BTD for equivalency.

approximations. Therefore the $i^{th}$ approximation of $\mathbf{X}$ is given as follows:

$$\mathbf{X} = \sum_{r=1}^{R_i} [(\mathbf{g}_r)_i + (\mathbf{G}'_r)_{i-1}] \times_1 [(\mathbf{b}_r)_i + (\mathbf{B}'_r)_{i-1}] \times_2 [(\mathbf{c}_r)_i + (\mathbf{C}'_r)_{i-1}]$$

$$\text{where} \begin{cases} \mathbf{g}, \ \mathbf{G}'_{(i-1)} \in \mathbb{R}^{t'_i \times u'_i \times v \times w} \\ \mathbf{b}, \ \mathbf{B}'_{(i-1)} \in \mathbb{R}^{t \times t'_i} \\ \mathbf{c}, \ \mathbf{C}'_{(i-1)} \in \mathbb{R}^{u \times u'_i} \end{cases} \tag{4}$$

where $\mathbf{g}$, $\mathbf{b}$ and $\mathbf{c}$ are the additional information from increased rank of approximation. $\mathbf{G}'$, $\mathbf{B}'$ and $\mathbf{C}'$ to represent total approximation from lower rank approximations in the form of cores and factors. However, the core or factor must be recomputed so that the dimensions match the ranks required $R_i$, which is not a trivial manipulation.

Weights for a grouped convolution can be "expanded": the expanded weight from a convolution with group-size $g$ can be used in a convolution with group-size $h$, where $h > g$, giving identical outputs:

$$\mathbf{W}_g *_g \mathbf{X} \equiv \Psi_{g \to h}(\mathbf{W}_g) *_h \mathbf{X} \tag{5}$$

where $\Psi_{g \to h}()$ is the expansion function, $\mathbf{W}_g$ is the weight for a grouped convolution, $*_g$ refers to convolution with group-size $g$, and $\mathbf{X}$ is the feature map on which the convolution is applied. This expansion allows us to conveniently reformulate the GroSS decomposition in terms of the successive convolutional weights obtained from BTD, rather than within the cores and factors directly. More specifically, we define the bottleneck weights for the $Nth$ order GroSS decomposition with group-sizes, $S = \{s_1, ..., s_N\}$, as follows:

$$\mathbf{R}_N = \Psi_{s_1 \to s_N}(\mathbf{R}_1) + \sum_{i=2}^{N} \Psi_{s_i \to s_N}(\mathbf{r}_i)$$

$$\mathbf{P}_N = \mathbf{P}_1 + \sum_{i=2}^{N} \mathbf{p}_i, \quad \mathbf{Q}_N = \mathbf{Q}_1 + \sum_{i=2}^{N} \mathbf{q}_i \tag{6}$$

$\mathbf{R}_1$, $\mathbf{P}_1$ and $\mathbf{Q}_1$ represent the weights obtained from the lowest rank decomposition present in the series. $\mathbf{r}_i$, $\mathbf{p}_i$ and $\mathbf{q}_i$ represent the additional information that the $i^{\text{th}}$ rank decomposition contribute to the bottleneck approximation:

$$\mathbf{p}_i = \mathbf{P}_i - \mathbf{P}_{(i-1)}, \quad \mathbf{r}_i = \mathbf{R}_i - \Psi_{s_{(i-1)} \to s_i}(\mathbf{R}_{(i-1)}), \quad \mathbf{q}_i = \mathbf{Q}_i - \mathbf{Q}_{(i-1)}. \tag{7}$$

This formulation involving only manipulation of the convolutional weights is exactly equivalent to forming the bottleneck components $\mathbf{r}_i$, $\mathbf{p}_i$ and $\mathbf{q}_i$ from $\mathbf{g}_i$, $\mathbf{b}_i$ and $\mathbf{c}_i$, as in the general BTD to bottleneck case.

Further, grouped convolution weight expansion, $\Psi()$, enables us to dynamically, and differentiably, change group-size of a convolution. In itself, this is not particularly useful; a convolution with a larger group-size is requires more operations and more memory, while yielding identical outputs. However, it allows for direct interaction between different ranked network decomposition and, therefore, the representation of one rank by the combination of lower ranks. Thus, GroSS treats the decomposition of the original convolution as the sum of successive order approximations, with each order contributing additional representation power.

### 3.3.1 TRAINING GROSS SIMULTANEOUSLY

The expression of a group-size $s_i$ decomposition as the combination of lower rank decompositions is useful because it enables the group-size to be dynamically changed during training. The expansion and summation of convolutional weights is differentiable and so training at a high rank, also optimises the lower rank approximations simultaneously. To the best of our knowledge GroSS is the first method that allows for the simultaneous training of varying group-size convolutions.

We leverage the series form of the factorisation during training, by randomly sampling a group-size for each decomposed layer at each iteration. We sample a group-size $s_i$ for each decomposed layer from the probability distribution:

$$p(s_i) = \frac{\exp(-\alpha i)}{\sum_{i=1}^{N} \exp(-\alpha i)} \tag{8}$$

where $s_i$ refers to the $i^{th}$ smallest decomposed group-size, $N$ denotes the number of available group-size and $\alpha$ is the sampling temperature. When $\alpha = 0$, $p(s_i)$ is a uniform distribution. By increasing the sampling temperature $\alpha$, we update the weights of lower group-sizes more aggressively, and implicitly enforce the lower order approximation to carries more of the signal of the approximation. We set the default sampling temperature to $\alpha = 4$ and provide experimental evaluation to justify this as an appropriate choice.

### 3.3.2 RESPONSE RECONSTRUCTION

The aim of factorising a convolutional layer is to ultimately mimic its performance on a particular task. However, the objective of the factorisation itself is to minimise the error between the original tensor and the approximation with respect to the Frobenius norm. While this goes some way to meeting the overall goal of similar performance, small errors in approximation can lead to drastic decreases in performance. Therefore, we encourage the decomposed layer to reconstruct the response of the original layer.

We follow the proposal of Jaderberg et al. (2014) were minimising the response approximation error is minimised through backpropagation. This is done by freezing all the standard (not decomposed) layers, and penalising the difference between the activation $\boldsymbol{A}'$ of decomposed layers and activation $\boldsymbol{A}$ of the original layer using the loss:

$$L_{RR} = \frac{\|\boldsymbol{A}' - \boldsymbol{A}\|_2}{\|\boldsymbol{A}\|_2} \tag{9}$$

Loss for response reconstruction is the Frobenius Norm of differences between activation of decomposed layer and the target activation, normalised by the Frobenius Norm of the target activation.

## 4 EXPERIMENTAL SETUP

In this section, we list the setup for our experimental evaluation. We first detail the dataset on which evaluation is conducted. Next, we describe the network architecture on which perform GroSS decomposition. Finally, we list the procedure for the decomposition, response reconstruction and finetuning.

### 4.1 DATASET

We perform our experimental evalutation on CIFAR-10 (Krizhevsky et al., 2014). It is a dataset consisting of 10 classes. The size of each image is $32 \times 32$. In total there are 60,000 images, which are split into 50,000 train images and 10,000 testing images. We further divide the training set into a training and validation splits with 40,000 and 10,000 images, respectively.

### 4.2 MODEL

We test on a 4-layer network with four convolutional layers, with channel dimensions of 32, 32, 64 and 64, followed by two fully-connected layers of size 256 and 10. The convolution layers all have

kernel dimensions $3 \times 3$, a bias term, stride of 1 and padding of 1. Each convolution is followed by a ReLU layer and $2 \times 2$ max-pooling. The first fully-connected layer has a ReLU applied to its output. Further, we use a dropout layer with dropout probability of 0.5 between the two fully-connected layers.

Convolutional weights in the network are initialised with the He initialisation (He et al., 2015) in the "fan out" mode with a ReLU non-linearity. The weights of the fully-connected layers are initialised with a zero-mean, 0.01-variance normal distribution. All bias terms in the network are initialised to 0.

The network is trained from scratch on CIFAR-10 training split for 100 epochs using stochastic gradient descent (SGD). We adopt a initial learning rate of 0.1 and momentum of 0.9. The learning rate is decayed by a factor of 0.1 after 50 and 75 epochs. We apply the following data normalisation and augmentation strategy to the training images: images are padded with 2 pixels and a random $32 \times 32$ crop is taken from the padded image; there is probability of 0.5 that the image will be horizontally flipped; all images are normalised to a mean of $(0.5, 0.5, 0.5)$ with variance $(0.225, 0.225, 0.225)$. We train the network 5 times and list the results of training in Table 2.

We select the full network with median accuracy and decompose all convolutional layers but the first. Group-sizes are set to all powers of 2 which do not exceed the number of in channels for that respective layer. Our formulation of GroSS decomposition as a series of convolutional weight differences (expanded weights in the case of the grouped convolution) means that we are able to use an off-the-shelf BTD framework (Kossaifi et al., 2019). For each group-size, we set the stopping criteria for BTD identically: when the decrease in approximation error between steps is below $1e^{-6}$, or $5e^5$ steps have elapsed. We define approximation error as the Frobenius norm between the original tensor and the product of the BTD cores and factors divided by the Frobenius Norm of the original tensor. Again, we perform this decomposition 5 times.

### 4.3 RESPONSE RECONSTRUCTION

Once decomposed, we perform response reconstruction on the decomposed layers simultaneously. The response reconstruction training lasts 30 epochs, which we found to be sufficient for convergence. Again, the response reconstruction stage is optimised through SGD with an initial learning rate was set to 0.0001, and momentum of 0.9. We decay the learning rate by a factor of 0.1 after 20 epochs. Inputs come from the CIFAR-10 train split. Target responses are generated by the full network. All parameters in the network aside from the decomposed pointwise layers and the grouped layers are frozen. The biases for the decomposed layers are also frozen.

### 4.4 FINETUNING

After we have performed response reconstruction on the factorised network, we then fine tune on the classification task. We tune for 150 epochs with an initial learning rate of 0.0001 and momentum 0.9. We decay the learning rate by a factor of 0.1 after both 80 and 120 epochs. Data augmentation remains the same as with training the full network. Once more, all network parameters are frozen aside from the GroSS decomposition weights.

### 4.5 BASELINE: FIXED CONFIGURATIONS

In addition to GroSS, we decompose the original network into 4 fixed configurations. These configurations simple with a single group-size selected for each decomposed layer: 1, 4, 16, and 32. They represent a baseline as a standard BTD network compression method, where these would be reasonable group-sizes with which to decompose the network. Importantly, they span almost the entirety of the possible performance envelopes available to our network: from the smallest depthwise compression, to nearly the largest. The accuracy and cost of these configurations is detailed in Table 2

We perform response reconstruction, followed by finetuning on each of these fixed configurations almost identically to the method outlined for our GroSS decomposition. However, the initial learning rates for response reconstruction and finetuning is set to 0.01 and 0.001, respectively. Also, the finetuning for these fixed configurations lasts 100 epochs, with the learning rate scaled by a factor

| Network | Median | Mean (std) | MACs (million) |
|---|---|---|---|
| Full | 83.88 | 83.99 (0.53) | 4.42 |
| Fixed - 32 | 83.70 | 83.69 (0.14) | 4.28 |
| Fixed - 16 | 82.84 | 82.83 (0.10) | 2.66 |
| Fixed - 4 | 82.09 | 82.06 (0.07) | 1.44 |
| Fixed - 1 | 81.76 | 81.74 (0.07) | 1.14 |

Table 2: Full network and fixed configuration accuracy and inference cost.

| Configuration | MACs (million) | Fixed | GroSS | After Train | $\Delta$ wrt Fixed |
|---|---|---|---|---|---|
| **16 16 16** | 2.66 | 82.83 (0.10) | 81.14 | - | - |
| 2 32 64 | 2.36 | - | 81.72 | 83.84 (0.12) | ↑ 1.00 |
| 8 16 64 | 2.51 | - | 81.60 | 83.83 (0.13) | ↑ 0.99 |
| **4 4 4** | 1.44 | 82.06 (0.07) | 80.37 | - | - |
| 2 4 8 | 1.33 | - | 80.90 | 82.76 (0.11) | ↑ 0.70 |
| 2 8 8 | 1.41 | - | 80.89 | 82.92 (0.09) | ↑ 0.86 |

Table 3: Exhaustive search. Here we evaluate the top 2 configurations returned from the exhaustive with the fixed configuration (bold) setting the upper bound for inference cost. We list the mean accuracy and standard deviation from 5 runs of the fixed configuration when trained separately, as well as the median accuracy of all configurations in the GroSS decomposition. After the new configurations have been trained separately, we detail whether they are still more accurate than the fixed configuration. The numbers in the configuration column correspond to the group-size of each decomposed layer within the network.

0.1 after 80 epochs. The schedule was reduced because the fixed configurations converged more quickly.

## 5 RESULTS

### 5.1 GROUP-SIZE SEARCH

Since we have trained the entirety of the group-size configurations of the decomposed networks simultaneously, we have effectively removed the train burden from architecture search. Therefore, to determine a candidate configuration, we evaluate all possible configurations. Specifically, we assign an the evaluation task of finding architectures which have higher accuracy, but lower inference cost than their respective baseline configuration. We choose the 4 and 16 fixed configurations. To do so, we simply filter any configuration with multiply accumulates (MACs) above the respective target configuration. After filtering, we can select the highest accuracy remaining.

Once a configuration has been selected, we decompose, response reconstruct and finetune exactly as described for the fixed configurations. This provides a fair comparison to the target configuration accuracy. The results of this search are shown in Table 3. The decomposition and tune is performed 5 times for each configuration and the mean and standard deviation are reported.

As can be seen in the results, the most accurate configurations for a particular performance bracket within the GroSS decompostion, remain more accurate when decomposed and finetuned individually. In all cases tested, a significant increase in accuracy was found in a configuration with cheaper inference. Another interesting note is that ascending group sizes along layers seems to be preferable. This is not necessarily intuitive and further emphasises the need for search among grouped architectures.

### 5.2 TRAINING SAMPLING STRATEGY

In this section, we evaluate the effect of the sampling distribution temperature on the performance profile of the model's search space. The aim is to create a profile which most accurately recreates that produced by separate decomposition and finetuning of the many possible group-size configurations. To evaluate this, In Figure 1, we show the performance profiles produced by differing values of

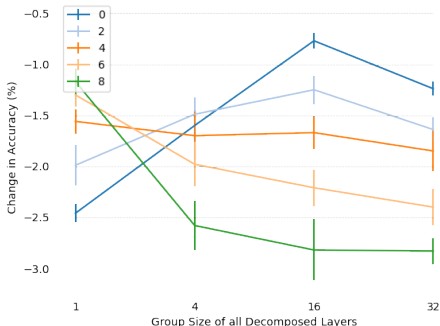 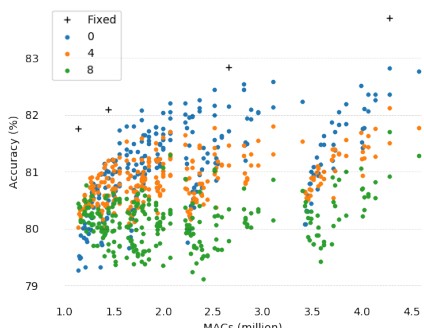

Figure 1: Sampling temperature's effect on accuracy profiles. **Left:** Here we evaluate how the sampling temperature impacts the accuracy of the GroSS decompostion at four configurations, which span the configuration space. We see that the accuracy deficit at $\alpha = 4$ remains relatively constant across the configurations, therefore providing the most balanced search space of temperatures tested. **Right:** We display the accuracy across all possible configurations. We see that $\alpha = 0$ performs well at the upper end of the performance envelope, whereas $\alpha = 8$ performs well with the smallest configurations. Temperature set at $\alpha = 4$ again provides the most balanced performance across all the configurations.

temperature $\alpha$ within our sampling distribution. We train each sampling temperature 5 times and plot the mean accuracy at a particular configuration. The percentage change in accuracy visualised is computed as the difference between the fixed configuration accuracy and the accuracy obtained from GroSS when running at the same group-size configuration.

As would follow intuition, higher temperatures are able to better recover accuracy for small group-size configurations, but lose significant accuracy at larger configurations. Conversely, low temperatures favour large group-sizes, but suffer with the smaller. Sampling with a temperature of $\alpha = 4$ provides the most balanced search space, as can be seen from its flat profile. We therefore use a this temperature setting as the default temperature in our finetuning stage.

## 6 CONCLUSIONS

In this paper, we have presented GroSS, a series BTD factorisation which allows for the dynamic assignment and simultaneous training of differing numbers of groups within a layer. We have demonstrated how GroSS-decomposed layers can be combined to train an entire grouped convolution search space at once. We demonstrate the value of these configurations through an exhaustive search, which is made possible through the simultaneous training. In doing this, we take a step towards shifting the burden of architecture search away from decomposition and training.

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
