# OpenReview forum: "GroSS Decomposition: Group-Size Series Decomposition for Whole Search-Space Training"
_ICLR.cc/2020/Conference — Reject_

### Official Review · AnonReviewer3 · 2019-10-23
**Official Blind Review #3**

**Rating:** 3

**Review:**

The authors propose to express the weight of a convolutional neural network as a coupled Tucker decomposition. The Tucker formulation allows for an efficient reformulation. The weigths of the sum of Tucker decompositions allows is randomly set at each iteration during training, with each term of the sum having a different rank.

The method is interesting, however the novelty is low. There is already large bodies of work on parametrizing neural networks with tensor decomposition, including coupled decomposition.


How does the proposed method compared to the related method DART? And to a simple coupled decomposition?
How is the method different to training several network with the same Tucker parametrization but different ranks? What about memory and computational efficiency?
In any case, these should be compared to.

The notation should be kept consistent throughout: e.g. either use t,u,v,w or d1,d2,d3,d4. Notation should be unified in the text and captions (e.g. Table 1).
In 3.2, when specifying the size of G, should it be G_r? Same for B and C.

Why the convolution R should be grouped? Should it not be a regular convolution?

For ot', u' being the group-size, what is o? It was not introduced.

The response reconstruction is only useful if the same, uncompressed network is already trained, would not be applicable for end-to-end training.

The model is a simple 4-layer network, not fully described. An established architecture, such as ResNet should be employed.

Experiments should be carried on ImageNet, or at least not just on CIFAR10.

There is no comparison with existing work, e.g. parametrization of the network with Tucker [1, 2] or CP [3]

[1] Compression of Deep Convolutional Neural Networks for Fast and Low Power Mobile Applications, ICLR 2016
[2] T-Net: Parametrizing Fully Convolutional Nets with a Single High-Order Tensor, CVPR 2019
[3] Speeding-up Convolutional Neural Networks Using Fine-tuned CP-Decomposition, ICLR 2015


**Experience Assessment:**

I have published in this field for several years.

**Review Assessment: Checking Correctness Of Derivations And Theory:**

I carefully checked the derivations and theory.

**Review Assessment: Checking Correctness Of Experiments:**

I assessed the sensibility of the experiments.

**Review Assessment: Thoroughness In Paper Reading:**

I read the paper at least twice and used my best judgement in assessing the paper.

---

> ### Author Response · Authors · 2019-11-11
> **Response to Reviewer #3 (1 of 2)**
>
> “The method is interesting, however the novelty is low. There is already large bodies of work on parametrizing neural networks with tensor decomposition, including coupled decomposition.”
>
> While there are works on parameterising and factorising networks with tensor decomposition, rank selection for decomposition remains relatively unexplored.
> The novelty of GroSS comes, not from the mechanics of decomposition, where we can use standard BTD due to the process described in Eq.6, but instead from the formulation of the search space as the combination (and interaction) of a number of series components. Each of these can be changed on-the-fly and therefore allow for simultaneous training. Here, we apply this series formulation to rank search for factorisation of networks, but we envisage similar series-based search can be used for a number of search tasks, such as number of channels and kernel dimensions.
>
> “How is the method different to training several network with the same Tucker parametrization but different ranks?“
>
> GroSS differs from the training of individual rank configurations since GroSS is able to train each rank factorisation of each layer, as well as all the possible combinations of ranks between layers. To explore our 4-layer network this would require 252 individual training runs, and 4^12 for our VGG16 network. However GroSS allows for all these configurations to be fine-tuned simultaneously, in a single training run.
>
> “Why the convolution R should be grouped? Should it not be a regular convolution?”
>
> We perform BTD, rather than pure Tucker decomposition. BTD is the extension of Tucker decomposition as it is the factorisation of a single tensor into the sum of multiple Tucker decompositions. When the number of Tuckers present in the BTD sum and the size of each Tucker kernel are set to the specific values, as described in our paper, the factorisation becomes equivalent to the grouped bottleneck architecture. Full derivation of this can be found in (paper ref: Yunpeng et al., 2017).
>
> “The model is a simple 4-layer network, not fully described. An established architecture, such as ResNet should be employed.”
>
> We agree and now have provided preliminary search results for VGG16 on CIFAR10, where we decompose all but the first convolutional layer into sizes [1, 4, 16, 32], therefore the number of possible configurations from our GroSS decomposition of VGG16 is 4^12. We implement a rudimentary breadth-first search to find configuration proposals. We can provide full implementation details of the search, network definition and fine-tuning strategy in an appendix.
>
> Config                                                       | MACs         | Accuracy
> -------------------------------------------------- | -------------- | ----------
> Baseline 4s                                              | 9.36M         | 91.00
> 1, 4, 1, 32, 1, 1, 16, 1, 4, 16, 32, 4           | 8.84M         | 91.28
> Baseline 16s                                            | 29.04M       | 91.48
> 1, 32, 16, 32, 16, 16, 32, 32, 32, 1, 4, 16 | 26.19M      | 91.56
> Full Network                                            | 313.74M    | 91.52
>
> Here, we perform similar testing to that with our 4-layer network. We set baseline configurations, where every layer is set to the same rank (4 or 16). We then employ GroSS to find a configuration which requires fewer MACs, while acheiving higher accuracy than the baseline. This was possible in each case, and notably one of our found configurations outperforms the original network before factorisation, requiring an order of magnitude fewer operations.

---

> > ### Author Response · Authors · 2019-11-11
> > **Response to Reviewer #3 (2 of 2)**
> >
> >
> > “There is no comparison with existing work, e.g. parametrization of the network with Tucker [1, 2] or CP [3]”
> >
> > Since we do not propose new mechanics or optimisation explicitly for the decomposition of the weight tensors, at any particular rank configuration the final result should be similar to other papers which employ BTD for network compression, which have been compared to other works. A contribution of GroSS, however, is that it allows search between these rank configurations. As such, we did not optimise fine-tuning or search strategy in favour of demonstrating that GroSS did enable search in the grouped convolution space.
> >
> > Therefore, GroSS should be compared with rank selection such as VBMF proposed in [4] (employed in [1]). We apply VBMF to our 4-layer network weights to estimate the rank of each layer, choosing the nearest value which satisfies the BTD to grouped bottleneck requirements. We then set the decomposition to this predicted rank (16, 8, 16), fine-tune and search, as described in the paper. The results are shown below:
> >
> > Config                     | MACs | Accuracy
> > ------------------------ | -------- | ------------
> > VBMF [4]: 16 8 16 | 2.51M | 83.33 (0.10)
> > 2 32 64 (in paper) | 2.36M | 83.84 (0.12)
> > 4 16 64                    | 2.22M | 83.93 (0.13)
> >
> > We are able to find lower cost configurations with improved accuracy over the VBMF estimation. In fact, in [1] they stated, “Although we can obtain very promising results with one-shot rank selection, it is not fully investigated yet whether the selected rank is really optimal or not.” Here we demonstrate that GroSS is a tool to capable of investigating this, and that VBMF is not optimal in our case.
> >
> > [4] Nakajima, Shinichi, et al. "Global analytic solution of fully-observed variational Bayesian matrix factorization." Journal of Machine Learning Research 14.Jan (2013).

---

### Official Review · AnonReviewer1 · 2019-10-24
**Official Blind Review #1**

**Rating:** 6

**Review:**

Summary: The authors introduce GROSS---a reformulation of block tensor decomposition, which allows multiple grouped convolutions (with varying group sizes) to be trained simultaneously. The basic idea is to reformulate the BTD so that higher-order decompositions can be expressed as functions of lower-order decompositions. Given this nesting, it is possible to implicitly train the lower-order decompositions while training the higher-order ones.

The authors frame this contribution as a form of "neural architecture search" (NAS), arguing that this allows researchers to simultaneously train grouped-convolution CNNs with varying group sizes. After the simultaneous training based on the GROSS approach, the researcher can then select the group size that gives the best performance/accuracy tradeoff. The selected model can be further fine-tuned on the task, and the authors found that this improved performance.

Empirical results on the CIFAR-10 dataset show that the proposed approach performs as expected, allowing for the simultaneous training of CNNs with grouped convolutions of varying orders. The results show the the proposed approach can find "better" solutions than a simple search over fixed architectures.

Assessment: Overall, this is a well-written and soundly derived contribution. However, it is quite niche and---while the authors frame it as a form of NAS---in my view, this contribution is more in the realm of hyperparameter search for grouped convolutions, and not NAS in general. I would recommend reframing the introduction to make this fact more explicit, as the approach does not provide a general strategy for differentiable NAS.  In addition, the empirical results are relatively shallow, with only one dataset and without detailed discussion of the variance of the results.

Reasons to accept:
- Well-written
- Sound and well-motivated algorithm
- Potential applications in cases where grouped convolutions are useful
- Empirical results demonstrate validity of the proposed approach

Reasons to reject:
- Relatively niche contribution incorrectly framed as general contribution to NAS
- Limited empirical analysis (e.g., only one dataset).

**Experience Assessment:**

I have read many papers in this area.

**Review Assessment: Checking Correctness Of Derivations And Theory:**

I assessed the sensibility of the derivations and theory.

**Review Assessment: Checking Correctness Of Experiments:**

I assessed the sensibility of the experiments.

**Review Assessment: Thoroughness In Paper Reading:**

I read the paper at least twice and used my best judgement in assessing the paper.

---

> ### Author Response · Authors · 2019-11-11
> **Response to Reviewer #1**
>
> “Overall, this is a well-written and soundly derived contribution. However, it is quite niche and---while the authors frame it as a form of NAS---in my view, this contribution is more in the realm of hyperparameter search for grouped convolutions, and not NAS in general. I would recommend reframing the introduction to make this fact more explicit, as the approach does not provide a general strategy for differentiable NAS.”
>
> While we agree that the specific method discussed in the paper is applied only to grouped convolutions search, we believe series-based representations of networks is a more general contribution to the architecture search task. We envisage a number of tasks where series-based search spaces can be trained, evaluate and searched using the philosophy as GroSS, such as number of channels present in a bottleneck layer or even kernel dimensions. We will aim to make the distinction between the series-based train and search philosophy, and our the specific application of GroSS for grouped convolution search.
>
> “In addition, the empirical results are relatively shallow, with only one dataset and without detailed discussion of the variance of the results.”
>
> We would like to point you towards the additional results presented in our response to Reviewer #3 for VGG16 on CIFAR10. We are continuing to work on more experiments, and hope to add them soon.

---

### Official Review · AnonReviewer2 · 2019-10-25
**Official Blind Review #2**

**Rating:** 3

**Review:**

Summary
---
This paper proposes to learn simultaneously all the parameters of grouped convolutions by factorizing the weights of the convolutions as a sum of lower rank tensors. This enables architecture search for the convolution parameters in a differentiable and efficient way.

Comments
---
I think was paper is well written, and was clear at least until 3.2. I believe some clarifications could be useful here, it is not written clearly that t' and u', the 2 first dimensions of the core are R times smaller than t and u. There is some explanation in the bracket (3) but 1) it should be stated clearly in the text, 2) I believe there are several typos on the 4th line of bracket (3) making it hard to understand.

I did not know about the expansion function, and while I trust the authors that it is correctly used, I would have like either more explanations on how it works or some reference.

Can you justify the softmax and the very high temperature? For N = 8, s_1 will be sampled 98.2% of the time s_2 1.8% and the other sampling probabilities are close to neglibigle. While I understand it seems to work better in practice, it looks extremely aggressive.

In 4.4 you say you perform finetuning for 150 epochs, which is huge, while on the abstract you said "GroSS represents a significant step towards
liberating network architecture search from the burden of training and finetuning". Can you comment?

As you say GroSS is an alternative to NAS (for the convolutions parameters that is), is the GroSS method proposed really faster and more accurate than a NAS baseline for finding these architectures?

I don't find the column titles in Table 3 to be always informative. "After train" means after the finetuning? I took me some time to realize the delta was the delta in accuracies, it is not very informative and it was not clear for me for some time what it meant. Either the titles should be chosen more carefully or the caption should be more precise I believe.

In figure 1, the legend should be more informative, at least incorporate a "alpha" or "temperature" title in the legend.

Conclusion
---
While the method is interesting I am wondering whether GroSS enables more efficient architecture search that tradional methods as there is still a long finetuning step, furthermore it can only be applied to grouped convolutions parameters. As the authors present it in the abstract and introduction as an alternative to NAS, I believe a comparison to a NAS would be needed.

**Experience Assessment:**

I do not know much about this area.

**Review Assessment: Checking Correctness Of Derivations And Theory:**

I assessed the sensibility of the derivations and theory.

**Review Assessment: Checking Correctness Of Experiments:**

I assessed the sensibility of the experiments.

**Review Assessment: Thoroughness In Paper Reading:**

I read the paper thoroughly.

---

> ### Author Response · Authors · 2019-11-11
> **Response to Reviewer #2**
>
> “I did not know about the expansion function, and while I trust the authors that it is correctly used, I would have like either more explanations on how it works or some reference.”
>
> As far as we are aware, we are the first to exploit the expansion of grouped convolution weights. We can provide more detail on the appearance of the expansion, however it is dependent on how the specific tensor/convolution library stores weights. Therefore, we hope to provide more intuition of how the expansion function works in the paper.
>
> “Can you justify the softmax and the very high temperature? For N = 8, s_1 will be sampled 98.2% of the time s_2 1.8% and the other sampling probabilities are close to neglibigle. While I understand it seems to work better in practice, it looks extremely aggressive.”
>
> When decomposing into multiple group sizes, each successive size in the series only aims to capture information not approximated by the previous order term. In Table 2, we show that even a depthwise factorisation of the network is able to recover almost all of the original accuracy (83.99 vs 81.74). Therefore, most of the energy of the approximation should be captured by the lowest rank term in the decomposition series. This provides intuition that the increasing rank terms in the series should be sampled with frequency that reflects the energy which they capture in the approximation, hence a high sampling temperature.
>
> “In 4.4 you say you perform finetuning for 150 epochs, which is huge, while on the abstract you said "GroSS represents a significant step towards liberating network architecture search from the burden of training and finetuning". Can you comment?”
>
> We found that for each single configuration, convergence was most reliably achieved by fine-tuning for 100 epochs. When a network is factorised using GroSS it is fine-tuned for a longer schedule of 150 epochs, but provides 252 configurations in our 4-layer network (4^12 configurations for our VGG16 decomposition). There are very likely more optimal fine-tuning strategies for both individual configurations and GroSS, but they still provide fair comparison to each other. If both training strategies were optimised, we would still expect to see the number of additional configurations trained by GroSS to vastly exceed the relative increase in number of epochs.

---

### Decision · Program_Chairs · 2019-12-19

**Decision:**

Reject

**Comment:**

The authors use a Tucker decomposition to represent the weights of a network, for efficient computation. The idea is natural, and preliminary results promising. The main concern was lack of empirical validation and comparisons. While the authors have provided partial additional results in the rebuttal, which is appreciated, a thorough set of experiments and comparisons would ideally be included in a new version of the paper, and then considered again in review.